# Burden of preconception morbidity in women of reproductive age from an urban setting in North India

**Ranadip Chowdhury[1], Sunita Taneja[1], Neeta Dhabhai[1], Sarmila Mazumder[1], Ravi Prakash Upadhyay[1], Sitanshi Sharma[1], Ananya Tupaki-Sreepurna[1], Rupali Dewan[2], Pratima Mittal[2], Harish Chellani[2], Rajiv Bahl[3], Maharaj Kishan Bhan[4], Nita Bhandari[1]***

**1** Centre for Health Research and Development, Society for Applied Studies, New Delhi, India, **2** Vardhman Mahavir Medical College and Safdarjung Hospital, New Delhi, India, **3** Department of Maternal, Newborn, Child and Adolescent Health, World Health Organization, Geneva, Switzerland, **4** Knowledge Integration and Translational Platform (KnIT), Biotechnology Industry Research Assistance Council (BIRAC), Department of Biotechnology, Government of India, New Delhi, India

* nita.bhandari@sas.org.in

**Data Availability Statement:** All relevant data are within the paper and its Supporting Information files.

## Abstract

### Background

There is a growing interest in the life course approach for the prevention, early detection and subsequent management of morbidity in women of reproductive age to ensure optimal health and nutrition when they enter pregnancy. Reliable estimates of such morbidities are lacking. We report the prevalence of health or nutrition-related morbidities, specifically, anemia, undernutrition, overweight and obesity, sexually transmitted infections (STIs) or reproductive tract infections (RTIs), diabetes or prediabetes, hypothyroidism, hypertension, and depressive symptoms, during the preconception period among women aged 18 to 30 years.

### Methods

A cross-sectional study was conducted among 2000 nonpregnant married women aged 18 to 30 years with no or one child who wished to have more children in two low- to middle-income urban neighborhoods in Delhi, India, in the context of a randomized controlled trial. STIs and RTIs were measured by symptoms and signs, blood pressure by a digital device, height by stadiometer and weight by a digital weighing scale. A blood specimen was taken to screen for anemia, diabetes, thyroid disorders and syphilis. Maternal depressive symptoms were assessed using the Patient Health Questionnaire-9 (PHQ-9). Multivariable logistic regression analysis was performed to identify sociodemographic factors associated with individual morbidity.

### Results

Overall, 58.7% of women were anemic; 16.5%, undernourished; 26%, overweight or obese; 13.2%, hypothyroid; and 10.5% with both symptoms and signs of STIs/RTIs. There was an increased risk of RTI/STI symptoms and signs in undernourished women and an increased

**Funding:** Initials of the authors who received each award: NB Grant numbers awarded to each author: BIRAC/GCI/0085/03/14-ACT) and Grant ID OPP1191052 The full name of each funder: Biotechnology Industry Research Assistance Council (BIRAC), Department of Biotechnology, Government of India under the Grand Challenges India-All Children Thriving initiative and the Bill & Melinda Gates Foundation, USA. URL of each funder website: https://birac.nic.in/; https://www.gatesfoundation.org/ The funders had no role in study design, data collection and analysis, decision to publish, or preparation of the manuscript.

**Competing interests:** The authors have declared that no competing interests exist.

risk of diabetes or prediabetes in overweight or obese women. An increased risk of undernutrition was also observed in women from lower categories of wealth quintiles. A decreased risk of moderate to severe anemia was seen in overweight women and those who completed at least secondary education.

## Conclusions

Our findings show a high burden of undernutrition, anemia, RTIs, hypothyroidism and prediabetes among women in the study. This information will aid policymakers in planning special programs for women of reproductive age.

## Introduction

Optimal health and nutritional status are essential for women of reproductive age. Early screening and treatment of morbidities are important to enable women to be prepared for future pregnancy [1, 2]. Anemia, under- or overnutrition, sexually transmitted infections (STIs) or reproductive tract infections (RTIs), hypothyroidism, diabetes and hypertension have been shown to be associated with an increased risk of adverse birth outcomes such as prematurity or small for gestational age (SGA) [3–5]. Specifically, anemia has been associated with the likelihood of having a baby born with low birth weight (LBW); underweight has been associated with LBW and preterm birth; smoking has been associated with preterm birth; and hypothyroidism has been associated with preterm birth, intrauterine growth restriction and LBW [3–5].

These associations have been largely based on observational studies. More recently, interventions delivered during prepregnancy and pregnancy, including additional food supplements for pregnant women with undernutrition, has been reported to increase birth weight and reduce SGA births [6]. The lack of planning for pregnancy and low rates of contraception use are of concern, as women are underprepared to enter pregnancy [7]. Detection and management of these morbidities through evidence-based intervention packages before pregnancy is important, as pregnancy is often reported several weeks post conception. Furthermore, some measures can be effective only when applied prior to conception, such as folic acid for the prevention of neural tube defects and adequate management of hypothyroidism [8, 9].

Reliable estimates of these morbidities in women of reproductive age are lacking in India.

We report the prevalence of anemia, undernutrition, overweight and obesity, STIs/RTIs, chronic diseases such as diabetes, prediabetes, hypothyroidism, and hypertension, and depressive symptoms among women aged 18 to 30 years residing in two urban neighborhoods in South Delhi, India. We also present associations of these morbidities with sociodemographic and economic factors.

## Materials and methods

### Subjects

A cross-sectional study among 2000 nonpregnant married women aged 18 to 30 years with no or one child who wished to have more children was conducted in two low- to middle-income urban neighborhoods in Delhi in the context of a randomized controlled trial (CTRI/2017/06/008908). In this trial, we conducted a door-to-door survey from July 1, 2017 to December 31, 2017 covering all households in Sangam Vihar and Dakshinpuri to identify 18- to 30-year-old

married nonpregnant women living with their husbands who had no or one child and wished to have more children. The poorest households (~20%) were excluded, as these households were without concrete roofing, toilets, water connection and legal electricity. Based on the population included, the results will be generalizable to most of the low- to middle-income population in urban India [10]. Women who did not give consent were excluded.

## Study procedures

Trained study workers assessed symptoms of STIs/RTIs. They measured blood pressure (Omron HBP 1300 digital blood pressure device; Omron Healthcare India), height (Seca-213 stadiometer) and clothed weight (Salter 9509 weighing scale) [11–13]. A nonfasting blood specimen was obtained from women who were allocated to the intervention group of the randomized controlled trial to screen for anemia (hemoglobin), diabetes (HbA1c), thyroid disorder [thyroid-stimulating hormone (TSH) and free thyroxine (FT4)] and syphilis (rapid plasma reagin; RPR).

Three milliliters of blood was collected in an evacuated tube containing ethylenediamine tetraacetic acid (EDTA; Becton Dickinson) to estimate the complete blood count (CBC) and HbA1c level, and 7 ml was collected in trace element-free serum-separating tubes. The sample was centrifuged at ~450 × g at room temperature for 10 minutes. The separated serum was transported in a cold box (4° to 8° C) to the Strand-Quest Diagnostics laboratory for TSH, ferritin, FT4 and RPR analyses [14].

The CBC parameters were measured on an LH 750 analyzer using AccuCount technology, an advanced analytical technique combining the Coulter principle of impedance counting and sizing with new sophisticated mathematical algorithms [15]. Serum ferritin was assessed using chemiluminescence in a Siemens Advia Centaur XP [16]. TSH and free T4 assays were performed using competitive immunoassay with direct chemiluminescence using acridinium ester technology [17]. HbA1c was assessed using the high-performance liquid chromatography technique in TOSOH-G8 [18]. The RPR test was performed by the flocculation technique [19].

## Definitions used

Anemia was defined as severe (hemoglobin: <8 g/dl), moderate (hemoglobin: 8–10.99 g/dl) or mild (hemoglobin: 11–11.99 g/dl) [20]. Low serum ferritin was defined by a cutoff of <30 ng/ml [21]. Undernutrition (BMI: <18.5 kg/m$^2$) was categorized as severe (BMI: <16 kg/m$^2$) or moderate (BMI: 16 to 18.49 kg/m$^2$). Overweight was defined as a BMI of 25 to 29.99 kg/m$^2$, and obesity was defined as a BMI ≥30 kg/m$^2$ [22]. Prediabetes was defined as HbA1c between 5.7% and 6.4%, and diabetes was defined as HbA1c ≥6.5% [23]. Women were treated with thyroxine if TSH levels were >5.5 IU/mL or if TSH levels were between 4.0 and 5.5 IU/mL and FT4 levels were <0.89 ng/dL [24]. Women were defined as stunted if their height was <150 cm [< -2 standard deviations (SDs) of the World Health Organization (WHO) standards; https://www.who.int/childgrowth/en/].

The symptoms of STIs/RTIs were assessed by trained study workers. An STI/RTI was considered if any one of the following symptoms were reported: swelling in the groin, dysuria, genital ulcer or sore, itching or burning sensation in the genital region, vaginal discharge, and pain in the lower abdomen. All women reporting one or more symptoms were examined by a physician, and an RTI was confirmed if any of the following signs were present: sores, blisters or ulcers in the genital area; foul-smelling, greenish or curdy white vaginal discharge; cervical erosion or mucopurulent pus at the cervical os; vaginal erythema with discharge; vulvar erythema, edema or induration; palpable lymph nodes in the inguinal area; and painful or

palpable adnexa on bimanual examination, lower abdominal tenderness, or cervical motion tenderness [25]. Syphilis was diagnosed by the RPR test [25].

Depressive symptoms were assessed using the PHQ-9 questionnaire validated for the Indian population [26]. Severe depressive symptoms were defined as PHQ-9 scores ≥15, and moderate depressive symptoms were defined as PHQ-9 scores between 10 and 14 [27].

Hypertension was based on physician confirmation of a systolic blood pressure ≥140 mm Hg and/or diastolic blood pressure ≥90 mm Hg [28]. If an initial blood pressure reading of ≥140/90 mm Hg was detected, a repeat measurement was taken after two days had passed. If the reading was still ≥140/90 mm Hg, the woman was referred to a tertiary care hospital (Safdarjung Hospital) for further assessment by a physician.

The wealth index was calculated for each participant by performing a principal component analysis based on all assets owned by the household, as done in national surveys [29]. The variables used were the source of drinking water; source of electricity; type of sanitation facility; type of cooking fuel used; construction material of the roof, floor and walls of the house; ownership of items such as a mattress, a pressure cooker, a chair, a cot/bed, a table, an electric fan, a radio/transistor, a black and white television, a color television, a sewing machine, a mobile telephone, any other telephone, a computer, a refrigerator, a watch or clock, a bicycle, a motorcycle or scooter, an animal-drawn cart, a car, a water pump, a thresher, a tractor and a house; number of household members sleeping in a room; and ownership of a bank or post-office account [29]. The total scores were used to divide the population into five equal wealth quintiles: the poorest, very poor, poor, less poor and the least poor.

## Ethical approvals

The ongoing trial is being conducted according to the guidelines outlined in the Declaration of Helsinki and is approved by the ethics committees of the Society for Applied Studies, New Delhi (SAS/ERC/LG/2017); Vardhman Mahavir Medical College and Safdarjung Hospital (IEC/SJH/VMMC/PROJECT-2017/694), New Delhi; and WHO, Geneva (ERC.0002934).

Written individual informed consent in the local language was obtained from the participants prior to enrollment. For those who were unable to read, the form was read aloud, and in those who were unable to sign, a thumb imprint was taken as witnessed by an impartial literate witness.

## Sample size

Sample size calculations were based on confidence interval using single proportion with relative precision. A sample size of 2000 women would enable us to measure the prevalence of specific morbidity (anemia, undernutrition, STIs/RTIs, hypothyroidism, overweight or obesity) with a relative precision ranging between 10% and 30%, at 5% alpha level and a nonresponse rate of 20%. Sample sizes table is provided in S1 Table.

## Statistical analysis

We used STATA version 15.1 (Stata Corporation, College Station, TX) for statistical analyses. Proportions and 95% confidence intervals (CI)) were calculated for categorical variables by binomial exact method, and means (SDs) or medians (interquartile ranges; IQRs) were calculated for continuous variables. One-way ANOVA was used to compare the mean serum ferritin values across anemia groups.

Multivariable logistic regression analysis was performed to identify baseline covariates (women's age, education, occupation and BMI; religion of the head of the family; wealth quintile; and family structure) independently associated with each morbidity (moderate to severe

anemia, hypothyroidism, undernutrition, overweight or obesity, prediabetes or diabetes, and symptoms and signs of RTI/STI). We included all baseline variables in each of the multivariable regression models. The multivariable logistic regression models were assessed for independence of observations, specification error, goodness-of-fit, multicollinearity, and influential observations [30]. We also used the "margins" command to calculate the predicted probability of each baseline variable holding all the other explanatory variables constant for each morbidity (S2 Table).

## Results

The baseline characteristics of the 2000 women included in this study are shown in Table 1. Approximately 35% of the women (692/2000) were stunted (height <150 cm). Nearly 80% (1632/1985) were Hindu by religion, and 45% (887/1985) belonged to the bottom two quintiles of the wealth index. The median (IQR) family income per year was 3333.3 (2500, 4300) USD, 90% of the households had bank accounts, and all enrolled households had concrete roofs and toilets, water connections within the house premises and legal electricity connections.

**Table 1. Baseline characteristics of surveyed participants.**

| Participant and Family/Household Characteristics | |
|---|---|
| | **n = 2000** |
| Age in years, mean (SD) | 24.1 (2.9) |
| Height in cm, mean (SD) | 152.3 (5.6) |
| Proportion of women with height | |
| <150 cm | 692 (34.6) |
| ≥150 cm | 1308 (65.4) |
| Years of schooling | **n = 1985** |
| Median (IQR) | 10 (8, 12) |
| Occupation | n = 1985 |
| Housewife | 1872 (94.3) |
| Working | 113 (5.7) |
| Religion of the head of the household | |
| Hindu | 1632 (82.2) |
| Muslim | 336 (16.9) |
| Others | 17 (0.9) |
| Family structure* | |
| Nuclear | 792 (39.9) |
| Extended or joint | 1194 (60.1) |
| Total family income per year (in USD) | |
| Median (IQR) | 3333.3 (2500, 4300) |
| Wealth quintiles | |
| Poorest | 382 (19.2) |
| Very poor | 505 (25.4) |
| Poor | 481 (24.2) |
| Less poor | 390 (19.7) |
| Least poor | 226 (11.4) |
| Any member of the household covered by a health scheme/health insurance | 226 (11.5) |

All values are number (percentages) unless stated otherwise

*Extended family: Family unit living with parents, their children and other dependent blood relatives; Joint family: parents and their male children with their families living together in a household.

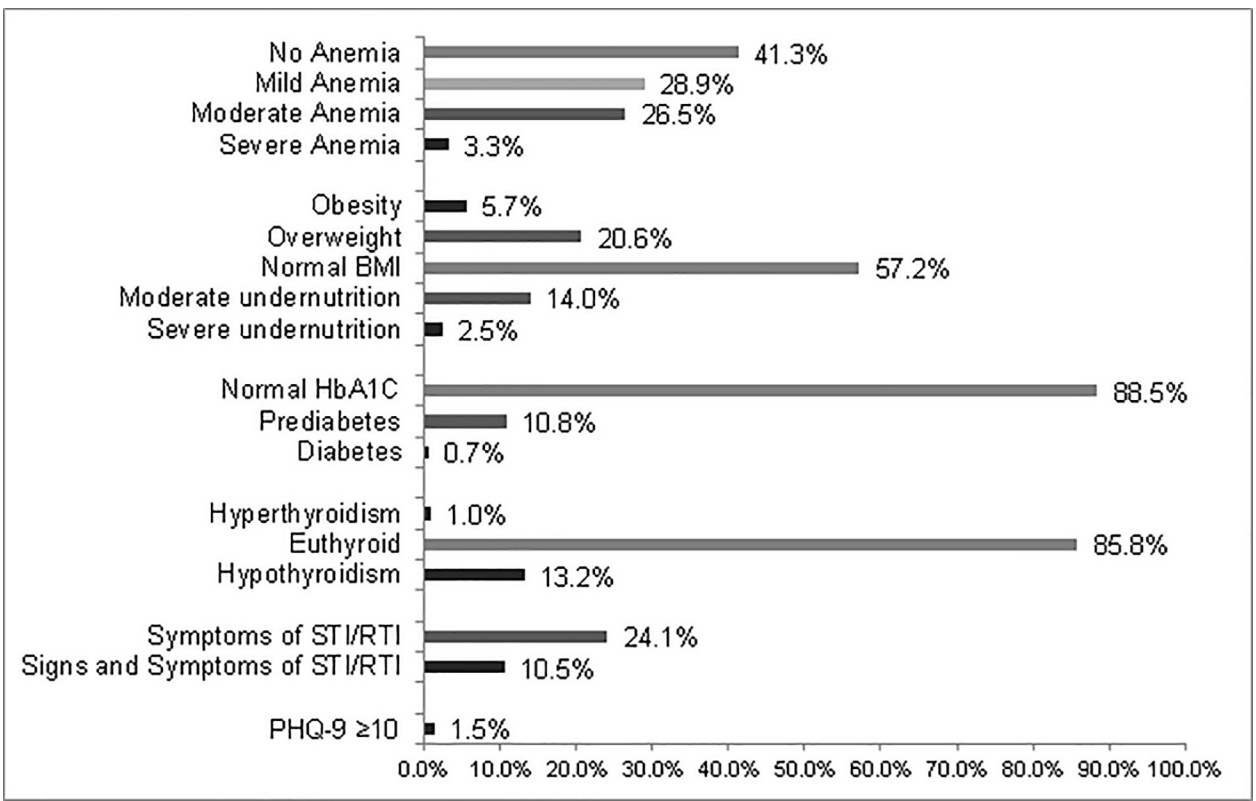

**Fig 1. Prevalence of morbidity among nonpregnant married women aged 18 to 30 years residing in low- to middle-income urban communities in Delhi.** Overall, 58.7% (95%CI: 56.5–60.9) of the women (1162/1979) had anemia, 16.5% (95%CI: 14.9–18.2) (330 women) were undernourished, and 26% (95%CI: 24.3–28.2) (525 women) were overweight or obese. In the 1979 women assessed, 13.2% (95%CI: 11.7–14.7) (261 women) were eligible for treatment with thyroxine. Prediabetes was detected in 10.8% (95%CI: 9.4–12.2) of participants, and only 0.7% (95%CI: 0.4–1.2) (14 women) had diabetes. Approximately one-tenth of the participants had both symptoms and signs of STIs/RTIs. In 29.9% (95%CI: 27.8–31.9) of the women, anemia was moderate to severe.

## Prevalence of morbidities

The prevalence of morbidities in the women who participated in the survey is summarized in Fig 1. Of the total women, 16% (95% CI: 14.4–17.7) did not have any previously listed morbidity, 84% (95% CI: 82.2–85.5) had one morbidity, 25% (95% CI: 23.1–27.0) had two morbidities and 5% (95% CI: 4.1–6.1) had 3 or more morbidities. Of the 2000 women, hemoglobin, TSH and HbA1c levels were ascertained in 98.9%; BMI, in all; symptoms of STIs/RTIs, in 99.9%; hypertension, in 99.3%; and PHQ-9 scores, in 96.1%.

The prevalence of morbidities that would require treatment under medical supervision in a hospital or at the primary care level is shown in Table 2. The former category included severe anemia, hypothyroidism, diabetes, hypertension, STIs/RTIs by syndromic diagnosis, undernutrition, obesity and severe depressive symptoms; this comprised 32.3% of the study participants. Nearly half of the women had a morbidity that could be managed at the primary care level.

Table 3 shows serum ferritin levels according to the severity of anemia. Notably, nearly all severely anemic women and over two-thirds of moderately anemic women had low serum ferritin levels. The proportion of anemic women with low serum ferritin levels was significantly higher among those with severe (98.5%; 64/65) and moderate (87.3%, 453/519) anemia than in those with mild anemia (69.3%, 389/561). There was a significant difference in the mean serum ferritin levels between the mild, moderate and severe anemia categories (ANOVA; p<0.001).

**Table 2. Prevalence of morbidities by severity and appropriate level for care among women aged 18 to 30 years residing in low- to middle-income urban communities in Delhi.**

| Morbidity | Prevalence n (%) | 95% CI |
|---|---|---|
| **Women with any severe morbidity requiring specialized medical supervision and follow-up** | 624 (32.3) | 30.2–34.4 |
| [met criteria for treatment with thyroxine (hypothyroidism) or had symptoms and signs of STIs/RTIs, severe anemia (Hb<8 g/dl), severe undernutrition (BMI <16 kg/m2), severe depressive symptoms (PHQ-9 score ≥15), diabetes (HbA1c ≥6.5%), confirmed hypertension, or obesity (BMI ≥30 kg/m$^2$)] | | |
| **Women with conditions that could be treated at primary level care** | 1139 (60.5) | 58.2–62.7 |
| [moderate anemia (Hb 8 to 10.99 g/dl), moderate undernutrition (BMI: 16 to 18.49 kg/m2), prediabetes (HbA1c: 5.7 to 6.4%), moderate depressive symptoms (PHQ-9 score: 10–14), or overweight (BMI: 25 to 29.99 kg/m$^2$)] | | |

Table 4 shows the association between individual morbidity and baseline characteristics from the multivariable logistic regression analysis. Increasing levels of schooling were protective against moderate to severe anemia; the odds of having moderate or severe anemia were 40% and 60% lower, respectively, among women who had completed secondary or higher levels of education than among those who had never been to school. The odds of having moderate to severe anemia were 35% lower in overweight women than in women with normal BMI. The odds of having hypothyroidism increased 5% with each year increase in the age of the women. There was a significant association between undernutrition and wealth quintiles with a dose response in the relationship. The odds of being overweight or obese increased 24% with each year increase in the age of the women. The odds of being overweight or obese were 85% and 60% higher among women who were in the top two wealth quintiles than among those who were in the lowest wealth quintile.

The risk of prediabetes or diabetes increased with overweight and obesity compared to normal weight. The odds of having prediabetes or diabetes increased by 12% with each year increase in the age of the women.

The odds of women having positive symptoms and signs of STIs/RTIs were 80% higher among women with BMI <18.5 kg/m$^2$ than among women with normal BMI (18.5 to 24.99 kg/m$^2$).

All multivariable logistic regression models were assessed for specification error, goodness-of-fit, multicollinearity and influential observations. We did not find any significant specification error and influential observations for any of the multivariable logistic regression models. Hosmer and Lemeshow goodness-of-fit statistics show that all models fit the data well. Variables included in the models did not show any collinearity. Details of the diagnostics for each model are provided in S3 Table.

**Table 3. Proportion of anemic women with low serum ferritin levels (<30 ng/ml).**

| Anemia | Serum ferritin | | Prevalence of low serum ferritin n (%) | 95% CI |
|---|---|---|---|---|
| | Mean (SD) | Median (IQR) | | |
| Severe anemia, n = 65 | 4.3 (4) | 4 (2–5) | 64 (98.5) | 91.7–99.9 |
| Moderate anemia, n = 519 | 16.3 (20.5) | 9 (6–18) | 453 (87.3) | 84.1–90.0 |
| Mild anemia, n = 561 | 28.1 (42.7) | 19 (10–33) | 389 (69.3) | 65.3–73.1 |

ANOVA; p<0.001

**Table 4. Multivariable analysis showing the association between baseline sociodemographic and anthropometry variables and individual morbidities.**

| | Adjusted OR (95% CI)* | | | | | |
|---|---|---|---|---|---|---|
| | Moderate to Severe Anemia | Hypothyroidism | Undernutrition | Overweight or Obesity | Prediabetes or Diabetes | Symptoms and Signs of STIs/RTIs |
| **Women's age** | 0.97 (0.93–1.01) | 1.05 (1.01–1.11) | 0.91 (0.87–0.95) | 1.24 (1.19–1.29) | 1.12 (1.06–1.18) | 1.01 (0.95–1.06) |
| **Women's years of schooling** | | | | | | |
| None (0) | Ref | Ref | Ref | Ref | Ref | Ref |
| Primary (1–5) | 0.78 (0.48–1.27) | 0.65 (0.35–1.24) | 1.13 (0.63–2.04) | 0.16 (0.64–2.10) | 1.20 (0.49–2.94) | 2.32 (0.98–5.50) |
| Secondary (6–12) | 0.58 (0.38–0.90) | 0.55 (0.32–0.94) | 0.92 (0.55–1.55) | 1.10 (0.64–1.80) | 1.09 (0.49–2.43) | 1.70 (0.76–3.80) |
| Higher than secondary (>12) | 0.40 (0.25–0.65) | 0.63 (0.35–1.14) | 0.81 (0.45–1.48) | 1.11 (0.64–1.92) | 1.31 (0.57–3.00) | 1.31 (0.56–3.09) |
| **Women's occupation** | | | | | | |
| Working | Ref | Ref | Ref | Ref | Ref | Ref |
| Housewife | 0.90 (0.58–1.37) | 0.71 (0.42–1.19) | 0.94 (0.55–1.63) | 0.79 (0.51–1.23) | 0.76 (0.43–1.34) | 0.56 (0.33–0.97) |
| **Religion of head of the household** | | | | | | |
| Others | Ref | Ref | Ref | Ref | Ref | Ref |
| Hindu | 1.30 (0.99–1.70) | 0.79 (0.56–1.11) | 1.14 (0.82–1.57) | 0.62 (0.47–0.82) | 1.02 (0.67–1.54) | 0.76 (0.53–1.09) |
| **Wealth quintiles** | | | | | | |
| Poorest | Ref | Ref | Ref | Ref | Ref | Ref |
| Very Poor | 0.93 (0.69–1.25) | 1.35 (0.89–2.07) | 0.67 (0.48–0.93) | 1.16 (0.82–1.65) | 1.15 (0.68–1.95) | 0.97 (0.62–1.52) |
| Poor | 1.05 (0.77–1.42) | 1.16 (0.74–1.81) | 0.64 (0.44–0.91) | 1.21 (0.85–1.72) | 1.37 (0.81–2.32) | 1.04 (0.66–1.66) |
| Less Poor | 0.90 (0.63–1.26) | 1.22 (0.76–1.99) | 0.49 (0.32–0.75) | 1.85 (1.28–2.69) | 1.78 (1.03–3.05) | 1.08 (0.65–1.79) |
| Least Poor | 0.98 (0.66–1.47) | 0.98 (0.55–1.73) | 0.40 (0.24–0.70) | 1.59 (1.04–2.43) | 1.74 (0.95–3.20) | 1.23 (0.69–2.20) |
| **Family structure** | | | | | | |
| Nuclear | Ref | Ref | Ref | Ref | Ref | Ref |
| Extended/Joint | 0.92 (0.74–1.14) | 1.04 (0.77–1.40) | 0.95 (0.73–1.24) | 0.84 (0.66–1.06) | 0.89 (0.64–1.23) | 1.20 (0.87–1.66) |
| **Women's BMI** | | | | | | |
| 18.5 to 24.99 kg/m$^2$ | Ref | Ref | | | Ref | Ref |
| <18.5 kg/m$^2$ | 1.29 (0.99–1.67) | 0.70 (0.46–1.07) | | | 0.76 (0.44–1.33) | 1.80 (1.24–2.60) |
| 25 to 29.99 kg/m$^2$ | 0.65 (0.49–0.86) | 0.99 (0.71–1.40) | | | 2.43 (1.72–3.45) | 1.09 (0.73–1.61) |
| ≥ 30 kg/m$^2$ | 0.67 (0.42–1.07) | 1.05 (0.60–1.84) | | | 9.29 (5.93–14.56) | 1.24 (0.66–2.31) |

*Estimates (OR) are shown for the variables included in the multivariable logistic regression model. For each condition, the comparator group was all women.

## Discussion

The salient finding of the study is that nearly half of the women of reproductive age in the study had a morbidity that could be managed within the primary health care system. Furthermore, in nearly one-third of the women, the nature of the morbidity was such that it would require management under careful medical supervision in a hospital setting. We observed an increased risk of RTI/STI symptoms and signs in undernourished women, diabetes or prediabetes in overweight or obese women and a decreased risk of moderate to severe anemia in overweight women. The relevance of these findings needs to be viewed in the context of the high burden of adverse birth outcomes such as preterm births and SGA babies in India and other low- and middle-income countries [31].

In our study, the high prevalence of hypothyroidism is of serious concern [24]. In human studies, hypothyroidism during early pregnancy has been associated with an increased risk of neurocognitive deficits in offspring, pregnancy loss and placental abruption [32]. Considering the high prevalence of hypothyroidism among women in the preconception period and its

adverse impact on pregnancy and fetal outcomes, timely screening of hypothyroidism before the planning of pregnancy and adequate management is critical. The observed prevalence of RTIs in ~10% of the women using the syndromic approach was lower than that observed in previous studies [33]. A previous study in a similar north Indian setting reported a 40% prevalence of RTIs in women aged 14–70 years [33]. Furthermore, etiological agents of RTI were detected in 12.5% of women with no symptoms [33]. The significance of asymptomatic colonization in the occurrence of adverse birth outcomes is unclear. We observed an increased risk of symptoms and signs of RTIs in undernourished women. Undernutrition may lower immune function and increase susceptibility to infection. The prevalence of prediabetes was similar to the findings from another study in India [34]. The link between maternal hyperglycemia and metabolic disease risk in offspring has been suggested in experimental models [35]. The clinical significance of a maternal prediabetic state on birth outcomes is uncertain and needs more research. Using a depressive symptom assessment tool to assess mental health status, 1.5% of women were classified as having moderate to severe depressive symptoms and in need of medical care. This was similar to the findings from the National Mental Health Survey (2015–16) in India, where the overall weighted prevalence of current depressive disorders in females older than 18 years was 2.97%, and in the age group of 18–29 years, it was 1.61% [36]. A national population-based survey showed that poor preconception mental health was associated with increased odds of experiencing any pregnancy complication and having LBW infants [37]. The data linking preconception mental health status and adverse birth outcome are limited.

In this study, the prevalence of anemia was similar to the national average reported in the National Family Health Survey; the prevalence of moderate to severe anemia was approximately 30% [29]. The serum ferritin findings in the anemic women in this study indicated that the vast majority of the moderately to severely anemic women had low serum ferritin and likely iron deficiency [38]. In the mildly anemic women, only half had low serum ferritin, indicating that other conditions may also account for anemia. In India, most anemic women receive weekly prophylaxis as a part of the national program rather than therapeutic doses of iron; this may be a cause of a persistence of high levels of anemia [39]. The risk of moderate to severe anemia was decreased in overweight women. Previous studies have also shown a decreased risk of anemia in overweight or obese women of reproductive age [40, 41]. Despite adiposity being protective against anemia, pregnant women who are overweight may have an increased risk of numerous other complications, including gestational diabetes, preeclampsia, pregnancy-induced hypertension, stillbirths, and preterm births [42]. Our study showed the dual burden of nutrition: 16.5% undernutrition and 25% overweight or obesity, which is similar to the national averages [29]. Prepregnancy underweight significantly increases the risk of preterm birth and SGA babies [4, 43]. Overweight, in particular obesity, may carry a higher risk of gestational diabetes and preterm birth [44]. Correction of aberrant nutritional status before conception seems important to reduce adverse birth outcomes and other consequences rather than initiation of corrective action only when pregnancy is confirmed.

Animal studies have shown that undernutrition or deficiency of specific nutrients and overnutrition can affect the embryo with a potential for future disease risk over their lifetime [2, 45–47]. Maternal overnutrition may cause defects in the mitochondrial phenotype, increased concentrations of inflammatory markers and chromosomal alterations in oocytes [2, 45–47]. Maternal undernutrition may reduce the concentration of circulating insulin and amino acids, which can lead to an altered growth trajectory of the fetus from before implantation [2, 45–47]. Disturbances in epigenetic mechanisms during early pregnancy may lead to an altered embryonic gene expression profile that persists through subsequent cell cycles and drives a modified developmental program [48].

The main strength of the current study was the careful ascertainment of morbidity using standard definitions and laboratory investigations performed in an internationally accredited laboratory. Our findings may be generalizable to 80% of similar populations of India [10]. Our estimate of women with both symptoms and signs of RTI was most likely an underestimate, as only 75% of those reporting symptoms agreed to a clinical examination.

There are several implications of the study findings. In countries such as India, special programs for adolescent health and nutrition have been initiated; however, their coverage and quality need improvement [49]. The adolescent health program ceases at 18 years of age, and consideration needs to be given to extend such a program to women up to 30 years of age. More research is required to estimate the prevalence of relevant health and nutrition disorders among women of reproductive age in India and the region. Health and nutrition disorders need to be managed prior to becoming pregnant and from early pregnancy to maximize the impact on birth outcomes. Improving primary health care, referral, and care at secondary and tertiary care hospitals are all important for the effective implementation of prepregnancy and pregnancy care in low- and middle-income countries.

## Conclusions

The present study demonstrates a high burden of several noncommunicable diseases, nutritional problems and infectious diseases in women of reproductive age during the preconception period, which have serious adverse effects on pregnancy and birth outcomes. For women to enter pregnancy anemia free, nutritionally replete and infection free, a preconception health program is of utmost importance.

## Supporting information

**S1 Table. Sample sizes.**
(DOCX)

**S2 Table. Predicted probabilities of baseline socio-demographic and anthropometry variables for Individual morbidities.**
(DOCX)

**S3 Table. Diagnostics of multivariable logistic regression models.**
(DOCX)

## Acknowledgments

The Society for Applied Studies acknowledges the core support provided by the Department of Maternal, Newborn, Child and Adolescent Health, World Health Organization, Geneva (WHO Collaborating Centre IND-158) and the Centre for Intervention Science in Maternal and Child Health (RCN Project No. 223269), Centre for International Health, University of Bergen (Norway). We are thankful to Dr. Nikhil Tandon (Professor and Head, Department of Endocrinology and Metabolism, All India Institute of Medical Sciences, New Delhi), Dr. Anju Virmani (Senior Consultant Endocrinologist) and Dr. Navin Dang (Senior Consultant Microbiologist) for their technical advice on diagnostic criteria and management of thyroid disorders. We are also thankful to Dr. Deepak More for supervising the quality assurance for the different assays.

All authors approved the final version of the manuscript.

## Author Contributions

**Conceptualization:** Ranadip Chowdhury, Sunita Taneja, Neeta Dhabhai, Sarmila Mazumder, Ravi Prakash Upadhyay, Sitanshi Sharma, Rajiv Bahl, Maharaj Kishan Bhan, Nita Bhandari.

**Data curation:** Ranadip Chowdhury, Sunita Taneja, Rajiv Bahl.

**Formal analysis:** Ranadip Chowdhury, Sunita Taneja, Rajiv Bahl.

**Funding acquisition:** Nita Bhandari.

**Writing – original draft:** Ranadip Chowdhury, Maharaj Kishan Bhan, Nita Bhandari.

**Writing – review & editing:** Ranadip Chowdhury, Sunita Taneja, Neeta Dhabhai, Sarmila Mazumder, Ravi Prakash Upadhyay, Sitanshi Sharma, Ananya Tupaki-Sreepurna, Rupali Dewan, Pratima Mittal, Harish Chellani, Rajiv Bahl, Maharaj Kishan Bhan, Nita Bhandari.

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
