## [Decision Letter · Decision Letter 0]

27 Nov 2019

PONE-D-19-26942

Burden of preconception morbidity in women of reproductive age from an urban setting in North India

PLOS ONE

Dear Dr. Bhandari,

Thank you for submitting your manuscript to PLOS ONE. After careful consideration, we feel that it has merit but does not fully meet PLOS ONE’s publication criteria as it currently stands. Therefore, we invite you to submit a revised version of the manuscript that addresses the points raised during the review process.

This study reports on the prevalence of morbidities/factors impacting preconception health in an urban setting in North India. Taking this life course approach to preconception health is of critical importance. The study is overall well conducted and reported. The reviewers have highlighted specific aspects of concern. Please address all reviewer comments in your revision. Please also ensure that your manuscript is thoroughly proof read for grammatical and spelling errors as, if your manuscript is accepted, PLOS ONE does not provide a detailed copy editing service.

We would appreciate receiving your revised manuscript by Jan 11 2020 11:59PM. To enhance the reproducibility of your results, we recommend that if applicable you deposit your laboratory protocols in protocols.io, where a protocol can be assigned its own identifier (DOI) such that it can be cited independently in the future. For instructions see: http://journals.plos.org/plosone/s/submission-guidelines#loc-laboratory-protocols

We look forward to receiving your revised manuscript.

Kind regards,

Briony Hill

Academic Editor

PLOS ONE

Journal Requirements:

1. We note that you have indicated that data from this study are available upon request. PLOS only allows data to be available upon request if there are legal or ethical restrictions on sharing data publicly. For information on unacceptable data access restrictions, please see http://journals.plos.org/plosone/s/data-availability#loc-unacceptable-data-access-restrictions.

Reviewers' comments:

Reviewer's Responses to Questions

**Comments to the Author**

1. Is the manuscript technically sound, and do the data support the conclusions?

Reviewer #1: Yes

Reviewer #2: Yes

2. Has the statistical analysis been performed appropriately and rigorously? 

Reviewer #1: Yes

Reviewer #2: Yes

3. Have the authors made all data underlying the findings in their manuscript fully available?

Reviewer #1: Yes

Reviewer #2: No

4. Is the manuscript presented in an intelligible fashion and written in standard English?

Reviewer #1: Yes

Reviewer #2: Yes

5. Review Comments to the Author

Reviewer #1: This paper uses cross sectional data on 2,000 married women 18 to 30 years of age selected from two low-med socioeconomic urban neighborhoods in Delhi, India to study health and nutrition related morbidities that are related to poor birth outcomes. The descriptive results show high rates of undernutrition, anemia, reproductive tract infections, hypothyroidism, and prediabetes. Multivariate results using logistic regression demonstrate associations of these morbidities to a basic set of variables such as age, education, and wealth.

On the whole, this is a well done study and I only have minor comments:

1. Women who did not give consent were excluded. It would be good to know the how many women were excluded and any information on how they may differ from the analysis sample.

2. In the multivariate analysis, the authors only included baseline variables that had a p value less than .20 in bivariate analyses. This is typically not a good idea since the baseline variables are likely correlated and omitting variables could lead to biased effects for the included variables. A better strategy would be to use the complete set of baseline variables in all regressions.

3. The coefficients in a logistic regression are scaled by an unknown factor. As a result, marginal effects which are not scale dependent are also often reported. Marginal effects are straightforward to calculate using the STATA statistical software that the authors used.

Reviewer #2: Preconception health and nutrition are recognised to impact on maternal and birth outcomes and an understanding of the prevalence of maternal health conditions and sociodemographic correlates is therefore important and understudied in this setting. This is a generally well designed, conducted and written study. Please perform a thorough proof-read to improve readability and punctuation. For example ‘planning special programs for the women’, ‘was such that it would require to be managed ‘ etc. There is some inconsistency in reporting of outcomes (eg variable inclusion or exclusion of overnutrition) and I would clarify this throughout the manuscript. The discussion is somewhat simplistic and would benefit from restructuring.

Line 23- I would reword the abstract of ‘We report prevalence of health and nutrition-related morbidities specifically, anemia, undernutrition, overweight and obesity, sexually transmitted infections (STI) or reproductive tract infections (RTI), diabetes or prediabetes, hypothyroidism, hypertension, and depressive symptoms during the preconception period, among women aged 18 to 30 years’ to health OR nutrition-related as I’m unclear how STI and RTI’s are nutrition related?

Line 30 - As this is a subset of a randomised controlled trial, the generalisability of the population needs to be commented on as these women would have volunteered for recruitment to a much more intensive research study and hence would be assumed to be more motivated.

Line 39 - ‘Significant associations were observed for RTI/STI and undernutrition, hypothyroidism and diabetes or prediabetes with increasing age, low body mass index (BMI) and lower wealth quintiles. Anemia was inversely

associated with women’s years of schooling.’ Please format appropriately to make the dependent and independent variables clearer.

Line 53 – This is worded a little clumsily (eg ‘health-related morbidity’). Suggest rephrase

Line 54-57 – I would provide further detail here about exactly what morbidities are associated with exactly what outcomes.

Line 58-60 – I would distinguish here between the need for interventions during pregnancy to both manage under and over nutrition.

Line 66 – ‘Reliable estimates of these morbidities in women of reproductive age are lacking in India.’ Is this the main research gap of this study? I would state this more clearly in the introduction and discussion.

Line 81 – Provide data to support this in the results (eg Table 1).

Line 85 – Provide more detail on the signs and symptoms of STI/RTI assessed and how (eg clinical checklist based on what)?.

Line 86 – State if weight measured clothed/unclothed or fasting/non-fasting and if blood sample non-fasting.

Line 150 – Not all the morbidities mentioned in the abstract are in the sample size calculations (eg over nutrition). Please specify primary versus secondary outcomes (ideally a priori). Overnutrition is also mentioned in the discussion as an important morbidity despite lack of inclusion in these sample size calculations and regression models.

Line 159-160/Table 4- Again, not all outcomes are mentioned (eg over nutrition).

Line 161 – Please provide details of how the models were assessed for standard assumptions.

Line 167 – Provide definition of stunting in the methods.

Line 188 – From this sentence it sounds like overweight/obesity don’t warrant medical treatment, is this the case?

Provide data on the proportion of women who responded from the total eligible.

Figure 1 – Suggest provide in greyscale and 2D

Table 4 – Not all significant results are reported in the text (eg BMI and anaemia), I would clearly state all significant relationships and then explain these in more detail.

There are no line numbers in the discussion so it is difficult to make comments. The formatting of the discussion also makes it difficult to review as it’s difficult to see where new paragraphs sometimes commence. The discussion appears to be written in a way where there are numerous very short paragraphs without consistent description of the consistency of findings to prior research and potential mechanisms/implications of the findings. I would suggest restructuring to 5-6 larger paragraphs

‘Animal studies have shown that undernutrition or deficiency of specific nutrients and

physiological status including hyperglycemia can affect the embryo with a potential for their

future disease risk over their lifetime [2, 32-34]’ Please clarify mechanistically how different physiological states affect the embryo in different ways.

6. PLOS authors have the option to publish the peer review history of their article (what does this mean?). If published, this will include your full peer review and any attached files.

Reviewer #1: No

Reviewer #2: No

---

## [Author Response · Author response to Decision Letter 0]

9 Jan 2020

RESPONSES TO REVIEWERS 

1. We note that you have indicated that data from this study are available upon request. PLOS only allows data to be available upon request if there are legal or ethical restrictions on sharing data publicly. For information on unacceptable data access restrictions, please see http://journals.plos.org/plosone/s/data-availability#loc-unacceptable-data-access-restrictions.

We have shared the dataset.

We have uploaded the anonymized data set. 

REVIEWER #1

This paper uses cross sectional data on 2,000 married women 18 to 30 years of age selected from two low-med socioeconomic urban neighborhoods in Delhi, India to study health and nutrition related morbidities that are related to poor birth outcomes. The descriptive results show high rates of undernutrition, anemia, reproductive tract infections, hypothyroidism, and prediabetes. Multivariate results using logistic regression demonstrate associations of these morbidities to a basic set of variables such as age, education, and wealth.

On the whole, this is a well done study and I only have minor comments:

1. Women who did not give consent were excluded. It would be good to know the how many women were excluded and any information on how they may differ from the analysis sample.

Of the women who had no other exclusion criteria, 13% did not consent to participate in the study. Some characteristics of the enrolled and non-enrolled women are given below. 

Baseline characteristics of enrolled and non-enrolled women

Characteristics Enrolled women 

(n=2000) Non-enrolled women 

(n=869)

Height <150 cm, n (%) 34.6% 34.7%

Height (cm), Mean (SD) 152.3 (5.6) 151.7 (6.2)

Body Mass Index (kg/m2), Mean (SD) 22.6 (4.3) 22.9 (3.9)

Mid Upper Arm Circumference (cm), Mean (SD) 25.7 (3.7) 25.3 (3.1)

Women years of schooling, Median (IQR) 10 (8, 13) 10 (8, 12)

Total family income per year (INR), Median (IQR) 200000 (144000, 264000) 180000 (120000, 240000)

2. In the multivariate analysis, the authors only included baseline variables that had a p value less than .20 in bivariate analyses. This is typically not a good idea since the baseline variables are likely correlated and omitting variables could lead to biased effects for the included variables. A better strategy would be to use the complete set of baseline variables in all regressions.

We have rerun the multivariable regression models using all the baseline variables namely women’s age, years of schooling, occupation, religion, household wealth quintile, family structure, women’s BMI. We have updated Table 4 accordingly. No major changes were observed in the effect size and its 95% CI. (Table 4; Pages 16-17).

3. The coefficients in a logistic regression are scaled by an unknown factor. As a result, marginal effects which are not scale dependent are also often reported. Marginal effects are straightforward to calculate using the STATA statistical software that the authors used.

We agree that the coefficients in a logistic regression are scaled by an unknown factor specifically for categorical variables. We have used the following command in STATA “margins, dydx(*)” to calculate the predicted probability of each baseline variable holding all the other explanatory variables constant. We have provided the predicted probability of each baseline variable with each morbidity (moderate to severe anemia, hypothyroidism, undernutrition, overweight or obesity, prediabetes or diabetes and symptoms and signs of RTI/STI) in the Supplement Table 1. The inference from logistic regression models and margins model is the same. 

We have accordingly modified the Statistical analysis section (Lines 189-194).

REVIEWER #2

Preconception health and nutrition are recognised to impact on maternal and birth outcomes and an understanding of the prevalence of maternal health conditions and sociodemographic correlates is therefore important and understudied in this setting. This is a generally well designed, conducted and written study. 

Please perform a thorough proof-read to improve readability and punctuation. For example ‘planning special programs for the women’, ‘was such that it would require to be managed ‘ etc. 

Thank you this suggestion. We hope that the manuscript reads better now.

There is some inconsistency in reporting of outcomes (eg variable inclusion or exclusion of overnutrition) and I would clarify this throughout the manuscript. The discussion is somewhat simplistic and would benefit from restructuring.

We have now included overnutrition (overweight or obesity) as an outcome. We have also restructured the Discussion section and hope that it provides better clarity now.

Line 23- I would reword the abstract of ‘We report prevalence of health and nutrition-related morbidities specifically, anemia, undernutrition, overweight and obesity, sexually transmitted infections (STI) or reproductive tract infections (RTI), diabetes or prediabetes, hypothyroidism, hypertension, and depressive symptoms during the preconception period, among women aged 18 to 30 years’ to health OR nutrition-related as I’m unclear how STI and RTI’s are nutrition related?

Thank you. We have rephrased the sentence as “We report the prevalence of health or nutrition-related morbidities, specifically, anemia, undernutrition, overweight and obesity, sexually transmitted infections (STIs) or reproductive tract infections (RTIs), diabetes or prediabetes, hypothyroidism, hypertension, and depressive symptoms, during the preconception period among women aged 18 to 30 years” (Lines 63-67).

Line 30 - As this is a subset of a randomised controlled trial, the generalisability of the population needs to be commented on as these women would have volunteered for recruitment to a much more intensive research study and hence would be assumed to be more motivated.

We excluded women living in temporary housing as they are likely to be relocated by the government in the near future. However, height, BMI, MUAC, years of schooling, total family income per year between enrolled and excluded women were not very different (see the table on Baseline characteristics of enrolled and non-enrolled women under Reviewer 1, Point 1). We have commented on the generalizability of the population in the Discussion section (Lines 365).

Line 39 - ‘Significant associations were observed for RTI/STI and undernutrition, hypothyroidism and diabetes or prediabetes with increasing age, low body mass index (BMI) and lower wealth quintiles. Anemia was inversely associated with women’s years of schooling.’ Please format appropriately to make the dependent and independent variables clearer.

We have rephrased the statement (see below) to clearly indicate dependent and independent variables (Lines 40-45). We hope that the dependent and independent variables are clearer now.

“There was an increased risk of RTI/STI symptoms and signs in undernourished women and an increased risk of diabetes or prediabetes in overweight or obese women. An increased risk of undernutrition was also observed in women from lower categories of wealth quintiles. A decreased risk of moderate to severe anemia was seen in overweight women and those who completed at least secondary education. 

Line 53 – This is worded a little clumsily (eg ‘health-related morbidity’). Suggest rephrase

We have now rephrased this as: 

“Optimal health and nutritional status are essential for women of reproductive age. Early screening and treatment of morbidities are important to enable women to be prepared for future pregnancy” (Lines 60-63). 

Line 54-57 – I would provide further detail here about exactly what morbidities are associated with exactly what outcomes.

We have provided details on the outcomes and the associated morbidities (Lines 66-70).

“Specifically, anemia has been associated with the likelihood of having a baby born with low birth weight (LBW); underweight has been associated with LBW and preterm birth; smoking has been associated with preterm birth; and hypothyroidism has been associated with preterm birth, intrauterine growth restriction and LBW.” 

Line 58-60 – I would distinguish here between the need for interventions during pregnancy to both manage under and over nutrition.

We have specified that interventions during pregnancy that are needed to manage undernutrition (Lines71-74).

Line 66 – ‘Reliable estimates of these morbidities in women of reproductive age are lacking in India.’ Is this the main research gap of this study? I would state this more clearly in the introduction and discussion.

The main research gap in India is lack of reliable estimates of health or nutrition related morbidities specifically, anemia, undernutrition, overweight and obesity, sexually transmitted infections (STIs) or reproductive tract infections (RTIs), diabetes or prediabetes, hypothyroidism, hypertension, and depressive symptoms among women of reproductive age.

Line 81 – Provide data to support this in the results (eg Table 1).

We have provided the following information in the Results section (Lines 203-206).

The median (IQR) family income per year was 3333.3 (2500, 4300) USD, 90% of the households had bank accounts, and all enrolled households had concrete roofs and toilets, water connections within the house premises and legal electricity connections.

Line 85 – Provide more detail on the signs and symptoms of STI/RTI assessed and how (eg clinical checklist based on what)?.

We are following “National Guidelines on Prevention, Management and Control of Reproductive Tract Infections including Sexually Transmitted Infections”, Ministry of Health and Family Welfare, Government of India (http://naco.gov.in/sites/default/files/National_Guidelines_on_PMC_of_RTI_Including_STI%201.pdf) for assessing symptoms and signs of STI/RTI. The symptoms and signs of STI/RTI are described in the Definition section; these are also summarized below. (Lines 134-144). 

“The symptoms of STIs/RTIs were assessed by trained study workers. An STI/RTI was considered if any one of the following symptoms were reported: swelling in the groin, dysuria, genital ulcer or sore, itching or burning sensation in the genital region, vaginal discharge, and pain in the lower abdomen. All women reporting one or more symptoms were examined by a physician, and an RTI was confirmed if any of the following signs were present: sores, blisters or ulcers in the genital area; foul-smelling, greenish or curdy white vaginal discharge; cervical erosion or mucopurulent pus at the cervical os; vaginal erythema with discharge; vulvar erythema, edema or induration; palpable lymph nodes in the inguinal area; and painful or palpable adnexa on bimanual examination, lower abdominal tenderness, or cervical motion tenderness [25]. Syphilis was diagnosed by the RPR test.”

Line 86 – State if weight measured clothed/unclothed or fasting/non-fasting and if blood sample non-fasting.

The weight was measured with clothes on and non-fasting blood sample was drawn. These have been specified in the manuscript (Line 104). 

Line 150 – Not all the morbidities mentioned in the abstract are in the sample size calculations (eg over nutrition). Please specify primary versus secondary outcomes (ideally a priori). Overnutrition is also mentioned in the discussion as an important morbidity despite lack of inclusion in these sample size calculations and regression models.

We have now calculated the sample size for prevalence of overweight or obesity. Assuming, prevalence of overweight or obesity as ~25%, with 15% relative precision and 95% confidence level, 512 women of reproductive age group are required. 

We did multivariable logistic regression to examine the association between baseline covariates (women’s age, education, occupation and BMI; religion of the head of the family; wealth quintiles; family structure) with overweight or obesity. We have now included these estimates in Table 4 (Pages 16-17).

Line 159-160/Table 4- Again, not all outcomes are mentioned (eg over nutrition).

We have now included overnutrition (overweight or obesity) in Table 4 (Pages 16-17).

Line 161 – Please provide details of how the models were assessed for standard assumptions.

The multivariable logistic regression models were assessed for independency, specification error (“linktest” command in STATA to calculate linear predicted value (_hat) and linear predicted value squared (_hatsq) , goodness-of-fit (we used Hosmer and Lemeshow’s goodness-of-fit test), multicollinearity ( “collin” command in STATA), influential observations ( “predict” command in STATA). 

Line 167 – Provide definition of stunting in the methods.

We have provided the definition of stunting [height <150 cm (< -2 standard deviations of the World Health Organization standards; https://www.who.int/childgrowth/en/] in the Methods section (Lines 131-134).

Line 188 – From this sentence it sounds like overweight/obesity don’t warrant medical treatment, is this the case?

We have now included overweight in the definition of women with conditions that could be treated at primary level care and obesity in definition of women with any severe morbidity requiring specialized medical supervision and follow up. Accordingly, we have revised Table 2 (Table 2; Page 13).

Provide data on the proportion of women who responded from the total eligible.

Of the total eligible women who met the inclusion criteria, 3604 (85%) women were enrolled and 675 (13%) did not give consent. 

Figure 1 – Suggest provide in greyscale and 2D

We have modified Figure 1 as requested. 

Table 4 – Not all significant results are reported in the text (e.g. BMI and anaemia), I would clearly state all significant relationships and then explain these in more detail.

We have now reported all significant results in the text (Lines 250-261) and explained these in greater details in the Discussion section (Lines 292-293; Lines 300-302; and Lines 322-327). 

There are no line numbers in the discussion so it is difficult to make comments. The formatting of the discussion also makes it difficult to review as it’s difficult to see where new paragraphs sometimes commence. The discussion appears to be written in a way where there are numerous very short paragraphs without consistent description of the consistency of findings to prior research and potential mechanisms/implications of the findings. I would suggest restructuring to 5-6 larger paragraphs

We have restructured the Discussion into 6 paragraphs and added line numbers. 

‘Animal studies have shown that undernutrition or deficiency of specific nutrients and

physiological status including hyperglycemia can affect the embryo with a potential for their future disease risk over their lifetime [2, 32-34]’ Please clarify mechanistically how different physiological states affect the embryo in different ways.

We have now clarified the mechanisms how different physiological status affect the embryo in different ways (Line 352-360).

---

## [Decision Letter · Decision Letter 1]

19 Mar 2020

PONE-D-19-26942R1

Burden of preconception morbidity in women of reproductive age from an urban setting in North India

PLOS ONE

Dear Dr. Bhandari,

Thank you for re-submitting your manuscript to PLOS ONE. We feel that it has greatly improved, but would like to ask you to make some minor changes to the manuscript, to meet PLOS ONE’s publication criteria. Therefore, we invite you to submit a revised version of the manuscript that addresses the minor points raised below.

We would appreciate receiving your revised manuscript by May 03 2020 11:59PM. To enhance the reproducibility of your results, we recommend that if applicable you deposit your laboratory protocols in protocols.io, where a protocol can be assigned its own identifier (DOI) such that it can be cited independently in the future. For instructions see: http://journals.plos.org/plosone/s/submission-guidelines#loc-laboratory-protocols

We look forward to receiving your revised manuscript.

Kind regards,

Frank Wieringa, M.D., Ph.D.

Academic Editor

PLOS ONE

Reviewers' comments:

Reviewer's Responses to Questions

**Comments to the Author**

1. If the authors have adequately addressed your comments raised in a previous round of review and you feel that this manuscript is now acceptable for publication, you may indicate that here to bypass the “Comments to the Author” section, enter your conflict of interest statement in the “Confidential to Editor” section, and submit your "Accept" recommendation.

Reviewer #1: All comments have been addressed

Reviewer #3: (No Response)

2. Is the manuscript technically sound, and do the data support the conclusions?

Reviewer #1: Yes

Reviewer #3: Yes

3. Has the statistical analysis been performed appropriately and rigorously? 

Reviewer #1: Yes

Reviewer #3: Yes

4. Have the authors made all data underlying the findings in their manuscript fully available?

Reviewer #1: Yes

Reviewer #3: Yes

5. Is the manuscript presented in an intelligible fashion and written in standard English?

Reviewer #1: Yes

Reviewer #3: Yes

6. Review Comments to the Author

Reviewer #1: (No Response)

Reviewer #3: A cross-sectional study was conducted to report the prevalence of health or nutrition-related morbidities during the preconception period among women from North India. The study also aimed to predict health and nutrition-related outcomes. The prevalence of health and nutrition-related morbidities ranged from 10 to nearly 60%.

Minor revisions:

1- Indicate the date range women participated in the study.

2- Line 164: Provide a more comprehensive sample size calculation.

State and justify the study’s target sample size with a pre-study statistical power calculation. The power calculation should include: sample size, alpha level (indicating one or two-sided), minimal detectable difference and statistical testing method.

3- Line 170: State the method used to estimate the 95% CIs.

4- Table 1: Indicate when frequency (%) are represented.

5- Line 203-8 : Provide 95% confidence intervals associated with the incidence percentages.

6- Table 3: Clarify that the 95% CIs are associated with the prevalence.

7- Include labeling that clearly identifies the morbidity factors.

8- The method section indicates that "multivariable logistic regression models were assessed for independence of observations, specification error, goodness-of-fit, multicollinearity and influential observations." Provide a summary of these indicators for each of the multivariate logistic models. Include a general statement about the fit in the results section, and include the full details as supplemental material.

7. PLOS authors have the option to publish the peer review history of their article (what does this mean?). If published, this will include your full peer review and any attached files.

Reviewer #1: No

Reviewer #3: No

---

## [Author Response · Author response to Decision Letter 1]

29 Apr 2020

RESPONSES TO REVIEWERS COMMENTS

REVIEWER #3: 

A cross-sectional study was conducted to report the prevalence of health or nutrition-related morbidities during the preconception period among women from North India. The study also aimed to predict health and nutrition-related outcomes. The prevalence of health and nutrition-related morbidities ranged from 10 to nearly 60%.

Minor revisions

1- Indicate the date range women participated in the study.

The women were enrolled from July 1, 2017 to December 31, 2017 (Lines 86-87). 

2- Line 164: Provide a more comprehensive sample size calculation.

State and justify the study’s target sample size with a pre-study statistical power calculation. The power calculation should include: sample size, alpha level (indicating one or two-sided), minimal detectable difference and statistical testing method.

Sample size calculations were based on confidence interval using single proportion with relative precision. We calculated sample size assuming a relative precision ranging between 10 and 30%, at 5% alpha level and assuming a non-response rate of 20% (Table below). These details have now been added under the section on sample size (Page 8). The table of sample sizes has been provided as Supplemental Table 1. 

3- Line 170: State the method used to estimate the 95% CIs.

We used binomial exact method to estimate the 95% CIs. 

4- Table 1: Indicate when frequency (%) are represented

Apologies, we have now indicated when frequencies (%) are represented (Page 10; Line 193)

5- Line 203-8: Provide 95% confidence intervals associated with the incidence percentages.

We have now provided 95% confidence intervals associated with the incidence percentages. (Pages 10-11)

6- Table 3: Clarify that the 95% CIs are associated with the prevalence.

We agree. The 95% CIs are associated with the prevalence. 

7- Include labeling that clearly identifies the morbidity factors.

The following labeling was used for categorizing morbidities: 

• Anemia: Reference category: No or mild anemia; other category: Moderate to severe anemia 

• Hypothyroidism : Reference category: hypothyroidism absent ;other category: hypothyroidism present 

• Undernutrition: Reference category: BMI � 18.5 kg/m2 ;other category: BMI <18.5 kg/m2

• Overweight and obesity : Reference category: BMI < 25 kg/m2 ; other category: BMI �25 kg/m2

• Prediabetes or Diabetes : Reference category: HbA1c <5.7%; other category: HbA1c �5.7%

• Symptoms and Signs of STIs/RTIs : Reference category: women who had either symptoms and no signs present or no symptoms present ;other category: women who had both symptoms and signs present 

The baseline variables were labelled as;

• Women’s age : continuous variable 

• Women’s education : Reference category :None; category 1: primary (1-5 years);category 2: secondary (6-12 years);category 3: higher than secondary (>12 years)

• Women’s occupation : Reference category: working outside home ; other category :not working (housewife)

• Women’s BMI : Reference category: BMI 18.5 to 24.99 kg/m2 ;category 1: BMI <18.5 kg/m2; category 2: BMI 25 to 29.99 kg/m2 and category 3: BMI� 30 kg/m2 

• Religion of the head of the family : Reference category :others (religions than Hindu );other category Hindu

• Wealth quintiles: Reference category: poorest; category 1: very poor; category 2: poor; category 3: less poor; category 4: least poor

• Family structure : Reference category :nuclear families; other category extended/joint families 

8- The method section indicates that "multivariable logistic regression models were assessed for independence of observations, specification error, goodness-of-fit, multicollinearity and influential observations." Provide a summary of these indicators for each of the multivariate logistic models. Include a general statement about the fit in the results section, and include the full details as supplemental material.

We have described the fit of the models in the results section (Lines 248-253). This is reproduced below 

All multivariable logistic regression models were assessed for specification error, goodness-of-fit, multicollinearity and influential observations. We did not find any significant specification error and influential observations for any of the multivariable logistic regression models. Hosmer and Lemeshow goodness-of-fit statistics showed that all models fit the data well. Variables included in the models did not show any collinearity. Details of the diagnostics for each model are provided in Supplemental Table 2.

---

## [Editor Report · Decision Letter 2]

3 Jun 2020

Burden of preconception morbidity in women of reproductive age from an urban setting in North India

PONE-D-19-26942R2

Dear Dr. Bhandari,

We are pleased to inform you that your manuscript has been judged scientifically suitable for publication and will be formally accepted for publication once it complies with all outstanding technical requirements.

With kind regards,

Frank Wieringa, M.D., Ph.D.

Academic Editor

PLOS ONE
---

## [Editor Report · Acceptance letter]

9 Jun 2020

PONE-D-19-26942R2 

Burden of preconception morbidity in women of reproductive age from an urban setting in North India 

Dear Dr. Bhandari:

I'm pleased to inform you that your manuscript has been deemed suitable for publication in PLOS ONE. Congratulations! Your manuscript is now with our production department. 

Kind regards, 

on behalf of

Dr. Frank Wieringa 

Academic Editor

PLOS ONE